# Post-Encoding Stress Does Not Enhance Memory Consolidation: The Role of Cortisol and Testosterone Reactivity

**DOI:** 10.3390/brainsci10120995

**Published:** 2020-12-16

**Authors:** Vanesa Hidalgo, Carolina Villada, Alicia Salvador

**Affiliations:** 1Department of Psychology and Sociology, Area of Psychobiology, University of Zaragoza, 44003 Teruel, Spain; 2Department of Psychology, Division of Health Sciences, University of Guanajuato, Leon 37670, Mexico; c.villada@ugto.mx; 3Department of Psychobiology and IDOCAL, University of Valencia, 46010 Valencia, Spain; alicia.salvador@uv.es

**Keywords:** cortisol, testosterone, stress, consolidation memory, young people

## Abstract

In contrast to the large body of research on the effects of stress-induced cortisol on memory consolidation in young people, far less attention has been devoted to understanding the effects of stress-induced testosterone on this memory phase. This study examined the psychobiological (i.e., anxiety, cortisol, and testosterone) response to the Maastricht Acute Stress Test and its impact on free recall and recognition for emotional and neutral material. Thirty-seven healthy young men and women were exposed to a stress (MAST) or control task post-encoding, and 24 h later, they had to recall the material previously learned. Results indicated that the MAST increased anxiety and cortisol levels, but it did not significantly change the testosterone levels. Post-encoding MAST did not affect memory consolidation for emotional and neutral pictures. Interestingly, however, cortisol reactivity was negatively related to free recall for negative low-arousal pictures, whereas testosterone reactivity was positively related to free recall for negative-high arousal and total pictures. This study provides preliminary evidence about a different reactivity of testosterone and cortisol to the MAST as well as on their effects on consolidation. Our results suggest a different pattern of relationships between these steroid hormones and the arousal of the negative images.

## 1. Introduction

It is widely assumed that memory can be influenced by both chronic and acute stress. However, although chronic stress clearly impairs memory function, the direction of the effects of acute stress on memory performance is not clear, due to numerous factors related to the stressor, the memory function assessed, the material to be remembered, and the individuals (i.e., sex and age) considered [1]. Therefore, it is necessary to shed light on the impact of acute stress on memory, given the important consequences for individuals.

The vast majority of studies investigating acute stress effects on memory have used cortisol levels to assess the endocrine response to stress [2]. Cortisol is released as a consequence of hypothalamic-pituitary-adrenal axis (HPA-axis) activation [3]. It exerts its effects on memory because it can cross the blood-brain barrier and bind to glucocorticoid and mineralocorticoid receptors which are located in crucial brain areas for memory, such as the hippocampus, the prefrontal cortex, and the amygdala [4]. Thus, stress-induced cortisol changes in these areas will affect memory performance.

According to Roozendaal’s model, stress-induced cortisol may have differential effects on the various memory phases, improving consolidation while impairing retrieval performance in a clearly adaptive mechanism [5]. This model suggests that cortisol blocks memory retrieval processes in favor of consolidation processes, facilitating the consolidation of new important information related to the stressor that should be recalled in similar future situations. Supporting this theory, a large number of studies have investigated the effects of acute stress on memory consolidation, that is, when the stressor is applied shortly after learning (i.e., post-encoding stress), with improvements found in young people [6,7,8,9,10,11,12,13,14,15,16]. However, not all previous papers have found these results because impairing effects [17] or even non-effects [18,19,20,21] have also been reported. Most of the papers showing an enhancing effect of stress on memory consolidation have found this effect for emotional (specifically negative) material, but not for neutral material [6,9,11,12,13,15,19], although Preuss and Wolf [10] found enhanced recall for neutral material, but not for emotional (i.e., positive and negative) material. Furthermore, some studies have found a positive [8,9,10,11,12,13,20] or curvilinear [7,14] relationship between the cortisol response to the stressor and memory performance. Hence, it is necessary to clarify the stress-induced cortisol effects on consolidation memory performance, given its implication in future dangerous situations. The current study aims to address this issue.

In comparison with cortisol, little attention has been paid to the testosterone response to acute stress except in contexts related to competition and social stress. The release of testosterone is under the control of the hypothalamic-pituitary-gonadal axis (HPG-axis), and like the HPA-axis, the HPG-axis is affected by stress. A large body of evidence points to the idea that stressful situations lead to a decrease in testosterone due to temporary suppression of the reproductive system because it is not needed to cope with the stressor [22]. However, depending on the nature of the stressor, this response can differ. Thus, situations involving challenge, particularly related to social status, would elicit testosterone responses [22], as proposed by the “Challenge Hypothesis” [23,24]. In this regard, testosterone levels increase in response to competitive and other social stressors (for a review, see [25]), mainly those involving evaluation by others and, therefore, a challenge to the self and potential status change. Thus, a psychosocial stressor like the Trier Social Stress Test (TSST; [26]), which asks participants to demonstrate that they are the best candidates for a desirable position, is considered highly competitive [27], and there is evidence of testosterone increases related to this psychosocial stressor [28,29,30,31], although this does not always occur [32]. No significant testosterone changes have been reported in response to a combined laboratory stressor (mental arithmetic, Stroop-task, and public speaking) [33], a public oral presentation at a scientific conference [34], or a cognitive stressor [35]. Therefore, there is still no consensus about which acute stressors are able to induce a testosterone response. The current study aims to examine whether an acute laboratory stressor that combines physical and psychosocial stress, the Maastricht Acute Stress Test (MAST) [36], is able to provoke a testosterone response in young men and women.

In addition, like cortisol, testosterone can affect memory performance through androgen receptors distributed in brain regions related to this cognitive function, such as the hippocampus, amygdala, and prefrontal cortex [37,38]. In fact, effects such as maintaining neurogenesis and supporting synaptic density in the hippocampus have been attributed to testosterone [39]. Supporting this, several studies carried out with healthy older people have reported a positive relationship between testosterone and verbal memory performance [40,41,42,43,44]. However, testosterone levels have also been associated with poorer semantic and episodic memory performance in middle-aged and older women [43], and no relationship has been found between testosterone and declarative memory in middle-aged and older people [42,45,46]. Finally, [47] reported a negative relationship between the total testosterone level and verbal learning and memory, but this result was not found for free testosterone in men from 22 to 45 years old. Therefore, the testosterone-memory link remains unclear, especially in young people, due to the lack of studies with this age group.

Moreover, apart from the aforementioned evidence, to our knowledge, no previous studies have investigated the testosterone response to an acute stressor and its effects on memory performance. Only two studies have investigated the testosterone response to a cognitive stressor (i.e., Rey Verbal Learning Test) and its effects on memory [35,48]. These authors reported a curvilinear relationship between testosterone levels and verbal memory performance in middle-aged women who were caregivers of offspring with autism spectrum disorder [35] and children with eating disorders [48].

Hence, the current study aimed to examine the psychobiological (i.e., anxiety, cortisol, and testosterone) response to the MAST and its effects on memory consolidation in healthy young men and women. We chose the MAST because it combines typical aspects of physical (pain) and psychosocial (social evaluative threat, uncontrollability, and unpredictability) stressors, such as the Cold Pressor Test (CPT; [49]) and the TSST, respectively, two commonly used acute laboratory stressors.

Based on previous studies, we hypothesized that the MAST would elicit a robust psychobiological response, that is, higher anxiety and cortisol levels [50,51]. Regarding testosterone, we proposed that the stressor would also provoke a significant response [22]. In addition, we expected an enhancing effect of the MAST on consolidation performance, specifically for emotional material [2].

## 2. Materials and Methods

### 2.1. Participants

Forty-two individuals volunteered to participate. However, because five participants did not fully complete the experiment, the final sample was composed of 37 undergraduate students (21 men and 16 women) from 18 to 27 years old (M = 21.78, SD = 2.53). A priori calculation of the sample size indicated that 38 participants were enough to achieve a high power (0.9).

Participants were randomly assigned to a stress (8 men and 10 women) or control (13 men and 6 women) condition. There were no differences in age (stress = 21.94 and control = 21.63), socioeconomic status (stress = 6.61 and control = 6.26), or body mass index (stress = 22.28 and control = 23.56) between the two conditions (all *p* > 0.216). There were no differences in sex between the stress and control conditions (χ^2^ = 2.165, gl = 1, *p* = 0.141).

All of the participants were undergraduate students from different degree programs. The exclusion criteria were as follows: smoking more than 10 cigarettes a day; alcohol or other drug abuse; dental, visual, or hearing problems; presence of cardiovascular, endocrine, neurological, or psychiatric disease; using medication related to emotional or cognitive functioning or able to influence hormone levels (i.e., glucocorticoids, psychotropic substances, or sleep medications); presence of a stressful life event; and having been under general anesthesia at least once during the past year. All the women had been oral contraceptive users for at least six months to control hormonal changes related to the menstrual cycle.

### 2.2. Procedure

Each participant had to attend two sessions on two consecutive days: acquisition and memory testing sessions. All sessions were carried out between 15.00 and 17.00 h, and each participant attended the two sessions at the same hour. Participants were asked to follow some recommendations that were verified before each session started. These recommendations were: (i) maintain their general habits, (ii) sleep as long as usual, (iii) refrain from heavy physical activity the day before, (iv) not consume alcohol since the night before, (v) refrain from eating, smoking, or taking any stimulants such as coffee, cola, tea, or chocolate two hours prior to the sessions, and (vi) not brush their teeth one hour prior to the sessions. All participants signed the informed consent form and received 15 € as compensation for completing the experiment. The study was approved by the Ethics Research Committee of the Aragon Community (Code: PI18/054) and adhered to the Helsinki Declaration.

#### 2.2.1. Session 1: Acquisition Session

After participants arrived at the laboratory, one of the researchers measured their weight and height. After that, during the habituation phase, participants filled out a general questionnaire on sociodemographic data and the State Anxiety Inventory (STAI, pre-task). In addition, they provided the first saliva sample (−80 min), in order to obtain the hormonal baseline concentrations. Then, 80 color pictures extracted from the Spanish version [52] of the International Affective Picture System (IAPS; [53]) were presented to participants. These pictures were selected according to their valence and arousal (16 positive-high arousal, 16 positive-low arousal, 16 negative-high arousal, 16 negative-low arousal, and 16 neutral). Each picture was presented individually for 5 s on a screen, followed by a black screen for 15 s while participants rated its valence (from 1 = very positive to 9 = very negative) and arousal (from 1 = high to 9 = low arousal) using the Self-Assessment Manikin (SAM; [54]) scale. Next, participants received the instructions for the stress or control task procedure (depending on the condition to which they were randomly assigned), and they provided the second saliva sample (−5 min).

In the stress condition, 18 participants were exposed to the Maastricht Acute Stress Test (MAST; [36]). The MAST is a standardized laboratory stress task consisting of five trials where participants have to submerge one hand in cold water with a temperature of 3 °C for a variable length of time (from 60 to 90 s). Between the hand immersion trials, participants had to perform a complex arithmetic task that consisted of subtracting as quickly and accurately as possible. When participants subtracted too slowly or made a mistake, they received negative feedback from an evaluator and had to start over. The evaluator was a different person from the experimenter and the opposite sex of the participant. During the stress task, participants remained standing and were filmed with a video camera.

In the control condition, a total of 19 participants were exposed to a control task, which was very similar to the stress task except that the hand immersion was in warm water (36 °C) and the arithmetic task was very easy (count from 1 to 25). The control task was carried out by the experimenter, and participants did not receive any feedback and were not filmed. Immediately after the stress or control task, participants completed the STAI (post-task) and provided the third saliva sample (+10 min).

The learning and stress/control tasks were performed in the same room. However, after the learning task, in both conditions, the cold pressor machine was introduced, and, in the stress condition only, the video camera as well.

After the stress or control task, a recovery phase took place. During this phase, participants provided the fourth and final saliva sample (+25 min). All the participants were asked to return to the laboratory 24 h later, and they were not informed about the procedure for the following day.

#### 2.2.2. Session 2: Memory Testing Session

Twenty-four hours later, participants returned to the laboratory. After the habituation phase of 15 min, participants completed the free recall and recognition tasks. On the free recall task, participants had to write as many of the pictures they had seen the previous day as possible for 15 min. They had to provide a short but detailed description of each picture recalled. Two blinded and independent raters assessed each description and determined to what picture (if any) it corresponded. To assess interrater agreement, we calculated the Interclass Correlation Coefficient (ICC). Because the ICC average across the five categories of pictures was high (from 0.979 to 0.994), free recall outcomes were calculated as the mean of the two raters’ scores. On the recognition task, participants viewed 160 pictures (80 old and 80 new) individually on a screen. Each set of pictures was composed of 16 positive-high arousal, 16 positive-low arousal, 16 negative-high arousal, 16 negative-low arousal, and 16 neutral pictures. Here, participants had to respond “yes” or “no” to indicate whether they thought the picture had been presented in Session 1 or not. Recognition outcomes were calculated as the difference between the percentage of hits (total number of pictures correctly recognized) and the percentage of false alarms (total number of pictures incorrectly recognized), as in Hidalgo et al. [55] and Cornelisse et al. [56].

In addition, two saliva samples were collected immediately before the free recall test and after the recognition test (pre- and post-memory testing session), with a time interval of approximately one hour between samples. Finally, participants were debriefed.

### 2.3. Anxiety

The state anxiety of the participants was assessed using the Spanish version [57] of the STAI form S [58]. This questionnaire consists of 20 items (e.g., ‘I feel at ease’, ‘I feel upset’) that evaluate how participants feel at that particular moment. Participants completed it using a 4-point Likert scale ranging from 0 (not at all) to 3 (extremely). In the present study, Cronbach’s alphas were 0.75 (pre-STAI) and 0.94 (post-STAI).

### 2.4. Hormonal Determination

Six salivary samples were collected for each participant through passive drool to assess cortisol and testosterone levels. Participants were asked to deposit 5 mL of saliva in plastic vials, and the samples were frozen at −80 °C. Biochemical analyses were performed by the first author in the Laboratory of Social Cognitive Neuroscience (Faculty of Psychology at the University of Valencia).

All the samples were analyzed in duplicate using Salimetrics enzyme-immunoassay cortisol and testosterone kits (Salimetrics, State College, PA, USA). Assay sensitivity was 0.007 µg/dL and 1.0 pg/mL for the cortisol and testosterone kits, respectively. For each subject, all the samples were analyzed in the same trial. The mean inter- and intra-assay coefficients of variations were all below 10%.

### 2.5. Statistical Analyses

Seven hormone data (one for cortisol and six for testosterone) were missing because participants did not provide enough saliva. These missing data were imputed using the expectation-maximization method [58,59]. After testing the normality of the hormonal data with the Shapiro–Wilk test and verifying that they did not follow a normal distribution (all *p* < 0.005), the hormonal values were logarithmic-transformed.

Student’s *t*-tests were conducted to assess differences in the demographic variables between the stress and control conditions. Chi-square statistics were calculated to explore differences in the number of men and women in the two conditions. Mixed ANOVAs were performed to investigate the anxiety, cortisol, and testosterone response. Condition (stress vs. control) was included as between-subject factor. However, the within-subject factor (“time”) was different depending on the studied variable, as follows: (i) for anxiety: pre-task vs. post-task; (ii) for cortisol and testosterone in the acquisition session: baseline or −80, −5, +10, and +25 min; (iii) for cortisol and testosterone in the memory testing session: pre vs. post. To investigate the effect of the condition on the rating of the picture material, free recall, and recognition performance, mixed ANOVAs were performed. Thus, we added condition as between-subject factor and category (negative-high, negative-low, neutral, positive-high, and positive-low) as a within-subject factor. Finally, bivariate Pearson’s correlations were conducted between the delayed free recall or recognition outcomes and cortisol and testosterone reactivity to stress, calculated as the delta value max-mean of the baseline level (BL) and −5 min saliva samples.

The minimum power value obtained, calculated using G*Power [60], for the significant ANOVAs was 0.98. Greenhouse–Geisser was used when the requirement of sphericity in the ANOVA for repeated measures was violated. Post-hoc planned comparisons were performed using Bonferroni adjustments for the *p* values. The level of significance was <0.05. The results shown are means ± SEM. We used SPSS version 24.0 version to perform the statistical analyses. The values in the cortisol and testosterone figures represent raw values and not logarithmic-transformed values for easy interpretation of the figure.

## 3. Results

Table 1 shows the mixed ANOVAs performed.

### 3.1. Anxiety Self-Reported

Overall, participants in the stress condition had higher anxiety levels that those in the control condition. In addition, the condition × time (F (1, 33) = 31.67, *p* < 0.001, η^2^_p_ = 0.490) interaction was also significant. Thus, although there were no significant baseline differences between conditions (*p* = 0.087), participants in the stress condition had higher anxiety levels after the task than participants in the control condition (*p* < 0.001). Moreover, the stressor elicited anxiety increases (*p* < 0.001), but no differences were found in the control condition (*p* = 0.701) (Figure 1).

### 3.2. Cortisol Response

#### 3.2.1. Acquisition Session

Figure 2A shows the mean cortisol levels for the stress and control conditions during the acquisition session. Results showed that, in the stress condition, cortisol levels were higher than in the control condition. Baseline cortisol levels (*p* = 0.108) and cortisol levels immediately before the onset of the task (*p* = 0.596) were similar in the two conditions. However, in the stress condition, cortisol levels 10 and 25 min after the onset of the task were higher than in the control condition (both *p* < 0.001). In addition, in the stress condition, baseline cortisol levels were higher than the levels immediately before the MAST (*p* = 0.001). Then, cortisol levels increased immediately after the MAST (*p* < 0.001), and no differences were found between cortisol levels immediately and 25 min after the onset of the MAST (*p* = 0.325). Although cortisol levels were higher in the last saliva sample, they were similar to baseline (*p* = 0.190). For the control condition, cortisol levels decreased during the session, in accordance with the cortisol circadian rhythm.

#### 3.2.2. Memory Testing Session

Figure 2B shows the mean cortisol concentrations for both conditions before (pre) and after (post) memory testing. Results indicated that, according to the circadian pattern of cortisol secretion, cortisol concentrations were lower after memory testing than before it (*p* < 0.001).

### 3.3. Testosterone Response

#### 3.3.1. Acquisition Session

Figure 3A shows the mean testosterone levels for the stress and control conditions during the acquisition session. Regardless of the condition, participants showed similar testosterone levels at baseline and immediately before the task (*p* = 0.150). However, testosterone levels were higher immediately after the task than before the onset of the task (*p* < 0.001), but 25 min after the onset of the task, although testosterone levels started to decrease, they were similar to the levels 10 min after the onset of the task (*p* > 0.99).

#### 3.3.2. Memory Testing Session

Figure 3B shows the mean testosterone concentrations for both conditions before (pre) and after (post) memory testing. Results showed that testosterone levels were similar before and after the task (*p* = 0.847) and in both conditions (*p* = 0.715).

### 3.4. Ratings of Picture Material

#### 3.4.1. Valence

Results revealed that the negative-high (M = 7.82, SEM = 0.26) pictures were rated higher than the negative-low (M = 6.38, SEM = 0.21), neutral (M = 4.93, SEM = 0.19), positive-high (M = 2.59, SEM = 0.17), and positive-low pictures (M = 2.70, SEM = 0.16) (all *p* < 0.001). No differences were found between positive-high and positive-low pictures (*p* = 0.99).

#### 3.4.2. Arousal

As expected, negative-high (M = 2.87, SEM = 0.24) and positive-high pictures (M = 3.46, SEM = 0.24) were scored as more arousing than the negative-low (M = 4.95, SEM = 0.24), positive-low (M = 5.64, SEM = 0.26), and neutral pictures (M = 6.37, SEM = 0.28) (all *p* < 0.001). Moreover, the negative-low pictures were scored as more arousing than the neutral pictures (*p* < 0.001) and similar to the positive-low pictures (*p* = 0.190). No differences were found between the positive-low and neutral pictures’ arousal scores (*p* = 0.096). Arousal scores were similar in both conditions (*p* = 0.560).

### 3.5. Memory Performance

#### 3.5.1. Free Recall

Participants recalled the negative-high pictures more than the other four types of pictures (all *p* < 0.001). In addition, they recalled the positive-low pictures more than the negative-low and neutral pictures (both *p* < 0.017). No differences were found between the free recall of negative-low, positive-high, and neutral pictures (all *p* > 0.99) (Figure 4).

#### 3.5.2. Recognition

Results showed that the negative-low pictures were more recognized than the positive-high and positive-low pictures (both *p* < 0.034), and similar to the recognition of negative-high (*p* = 0.085) and neutral pictures (*p* = 0.999). In addition, the positive-high pictures were significantly less recognized than the negative-high, negative-low, and neutral pictures (all *p* < 0.036), and similar to the recognition of positive-low pictures (*p* = 0.074) (Figure 5).

### 3.6. The Relationship between the Hormonal Response and Consolidation Memory Performance

Results showed a negative relationship between cortisol reactivity and free recall for negative-low pictures (*r* = −0.486, *p* = 0.041). In addition, testosterone reactivity was positively related to free recall for negative-high (*r* = 0.586, *p* = 0.011) and total (*r* = 0.594, *p* = 0.009) pictures. However, no relationships were found between cortisol reactivity (all *p* > 0.169) and testosterone reactivity (all *p* > 0.148) and recognition performance for all categories of pictures (Figure 6).

## 4. Discussion

In the present study, we investigated the psychobiological response to the MAST and its influence on memory consolidation for emotional and neutral pictures in healthy young adults. The main results showed that the MAST increased the anxiety and cortisol levels, but it did not significantly change the testosterone levels. In addition, the post-encoding MAST did not affect memory consolidation for emotional and neutral pictures, assessed by free recall and recognition tasks 24 h after encoding. Finally, cortisol reactivity was negatively related to free recall for negative low arousal pictures, whereas testosterone reactivity was positively related to free recall for negative-high and total pictures. No associations were found between the reactivity of cortisol and testosterone and recognition performance.

As hypothesized, the MAST elicited a psychological and endocrine response in participants assigned to the stress condition. Specifically, although no differences in baseline levels of anxiety and cortisol were found between conditions, after the stress manipulation, participants in the stress condition showed higher anxiety scores and cortisol levels than participants in the control condition. These results are in line with previous studies reporting subjective (i.e., perceived stress and mood) ([36]; but see [61] for an absence of MAST effects on anxiety) and cortisol [36,61] responses to the MAST. Therefore, the stress manipulation (i.e., the MAST) was able to induce stress at both the psychological and physiological levels, as indicated in a previous review [50].

Although participants showed higher testosterone levels immediately and +25 min after the task than before the task, this time effect occurred regardless of the condition factor. Therefore, in contrast to the findings for cortisol, the MAST failed to provoke significant changes in the testosterone levels. This lack of stress effects on testosterone levels agrees with previous studies reporting no testosterone changes in some acute stressful situations [32,33,34,35,62]. This result could be explained by the possibility that our participants may not have perceived the MAST as a challenge, supporting the so-called challenge hypothesis [23,24]. According to this hypothesis, testosterone levels would increase when the individual is facing a situation where his/her social status is at stake. Testosterone increases in competitive and/or challenging contexts may be related to their functional meaning in aggressive, competitive, sexual, parental, and social behaviors [25]. Then, although the MAST combines aspects of physical and psychosocial stress, it is plausible, as Schoofs and Wolf [32] mention regarding the TSST, that the stressor was not severe enough to elicit a testosterone response. Finally, it is important to note that, although no condition differences were found, participants in the control condition seemed to have higher testosterone levels than those in the stress condition. This could be explained by the fact that the control condition was composed of twice as many men as women.

The lack of MAST effects on testosterone levels contradicts previous studies showing an increase in testosterone levels in response to the TSST only in young men [29] and in men and women [28,30,31]. However, in contrast to the current study, these studies did not include a control group or condition, which can explain this contradictory result. Further research should address the study of the testosterone response to acute stress, specifically to the MAST, because, to the best of our knowledge, this is the first study to investigate the salivary testosterone response to this laboratory stressor.

Unexpectedly, we failed to find stress effects on free recall for emotional and neutral pictures. This result is consistent with a few studies that failed to find stress effects [18,19,20]. However, it contrasts with most of the studies carried out on this issue, which reported enhancing effects on free recall [6,7,8,9,10,11,12,13,14,15,16] and impairing effects [17]. Several factors could be underlying this lack of a stress effect. For example, as mentioned above, Roozendaal’s model supports the idea that stress enhances consolidation but impairs retrieval [5]. However, the stress impact on consolidation is far more complex than this statement assumes. Some authors have reported an inverted-U pattern in cortisol’s effects on consolidation [7,14]. Thus, moderate cortisol levels facilitate memory consolidation, whereas low and high cortisol levels impair it. Therefore, it is conceivable that the MAST is a stressor that elicits a stronger cortisol response than the CPT [50], the stressor used in most of the studies that report enhancing stress effects [6,7,9,11,12,13,14,15,16]. In any case, all these issues should be taken into consideration in future research.

Although we did not find acute stress effects on memory consolidation, we reported a negative correlation between the cortisol reactivity to the stressor and free recall for negative-low pictures and, at the same time, a positive relationship between the testosterone reactivity to the stressor and free recall for negative-high and total pictures. Previous studies have shown positive relationships between cortisol reactivity to stress and memory consolidation for neutral material [8,10,18], for neutral material but not for negative material [11,13], for both negative and neutral material [9], and only for negative material in men and for neutral material in women [12]. The different direction of our correlation could be due to methodological differences such as the stressor task used, and therefore, the different cortisol magnitude elicited, and/or the different material and the number of stimuli to be remembered. Regarding the positive association between the testosterone reactivity to the MAST and free recall performance, no previous studies have investigated this relationship. Thus, this is the first time this association has been reported. This result coincides with previous authors who found positive associations between testosterone levels and verbal memory performance in healthy older people [40,41,42,43,44]. In addition, it supports previous findings that suggest that high testosterone levels could have neuroprotective effects when cortisol levels are high [63]. However, it is important to mention that these findings should be interpreted with caution, due to type 1 error, and only the correlations with testosterone reactivity are close to significance. It is possible that these associations would remain with a larger sample size. Further research should take these associations into consideration.

Consistent with the findings from the free recall task, no condition differences were found on the recognition task. Although post-encoding stress effects on recognition performance have been understudied compared to free recall, most studies investigating this issue failed to find stress effects [16,20,21], although Yonelinas et al. [19] found enhancing effects of stress on the recognition task. Interestingly, Sazma et al. [64] found different results depending on the context of the stressor. That is, when the stressor occurred in the same context as the study materials, the stressor enhanced the recognition performance. However, when the stressor occurred in a different context from the study materials, it did not affect recognition performance. In contrast to these findings, in the current study, the MAST took place in the same context as the learning material. In addition, Sazma et al. [64] and Shields et al. [65] found enhancing effects of stress on recognition, but only in recollection. These findings support the idea that post-encoding stress may only benefit recollection in recognition memory performance. It is possible that we failed to find stress effects on the recognition task because recollection was not considered in this study. Finally, and in contrast to the free recall task, no associations between cortisol and testosterone reactivities to stress and the recognition task were found. Perhaps, unlike free recall, recognition is not sensitive to the cortisol and testosterone levels, given that it requires a different mental process from the free recall test, it is easier, and performance is higher than on free recall, leading to a possible ceiling effect, which could explain this lack of a relationship with the endocrine response. Thus, more research is warranted to clarify all these aspects.

Although this is the first study to investigate the testosterone response to the MAST and its effects on memory consolidation, it is important to mention a few limitations. First, all the women included in the current study were oral contraceptive users. It is well known that not only the sex factor, but also the phase of the menstrual cycle, is a key factor when examining the effects of stress on memory performance, as previous reviews state [2,66]. Therefore, it would be interesting to incorporate not only oral contraceptive users, but also other groups of women in different phases of the menstrual cycle, such as follicular and luteal, in order to draw a complete picture of the stress effects and, specifically, the MAST’s effects on memory consolidation in young people. Second, further research should balance the number of men and women in the two conditions, in order to study the possible sex differences in the effects of stress on memory consolidation.

## 5. Conclusions

In sum, the present experiment suggests that, in healthy young people, the Maastricht Acute Stress Test, a laboratory stressor that combines physical and psychosocial stress components, is able to evoke a psychological (i.e., anxiety) and physiological (i.e., cortisol) response. However, it does not seem to significantly affect testosterone levels and memory consolidation. In addition, depending on the hormone, cortisol or testosterone, the endocrine reactivity has a different association with memory consolidation. These results should be considered preliminary evidence and analyzed more in depth.

## Figures and Tables

**Figure 1 brainsci-10-00995-f001:**
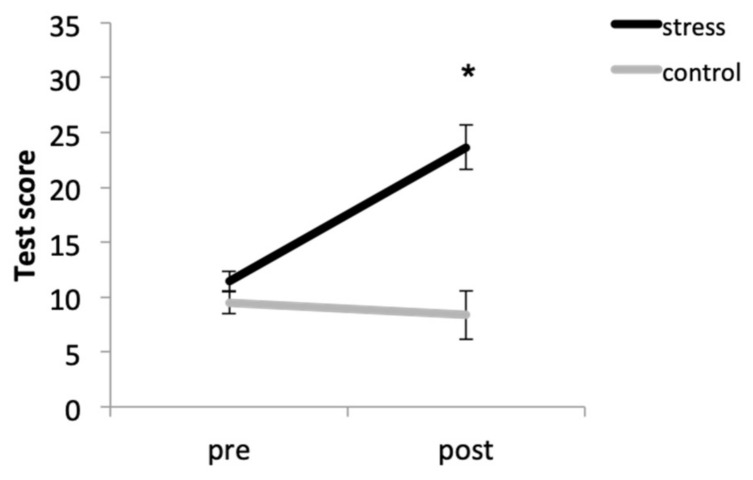
Anxiety scores before (pre) and after (post) the stress or control tasks (* *p* < 0.001).

**Figure 2 brainsci-10-00995-f002:**
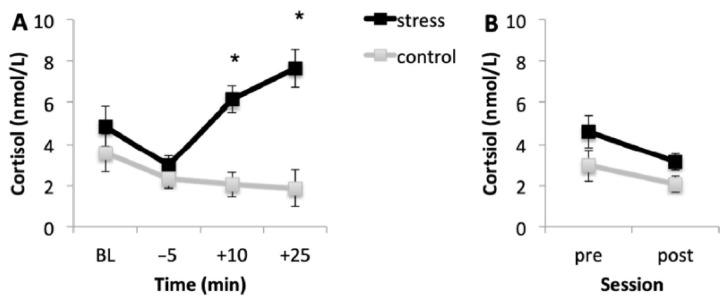
Salivary cortisol levels in the stress and control conditions for the acquisition session (**A**) and for the memory testing session (**B**). Error bars represent standard error of mean (S.E.M.). BL = baseline levels (−80 min) (* *p* < 0.001).

**Figure 3 brainsci-10-00995-f003:**
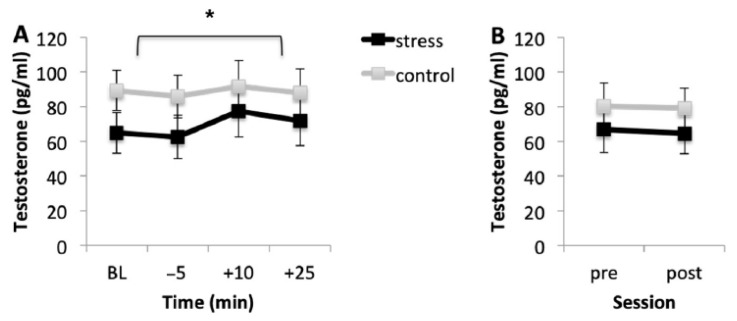
Salivary testosterone levels in the stress and control conditions for the acquisition session (**A**) and for the memory testing session (**B**). Error bars represent standard error of mean (S.E.M.). BL = baseline levels (−80 min) (* *p* < 0.001).

**Figure 4 brainsci-10-00995-f004:**
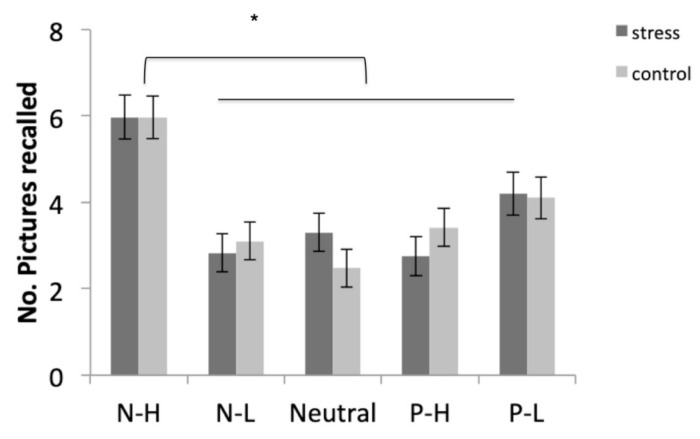
Free recall performance for negative-high (N-H), negative-low (N-L), neutral, positive-high (P-H), and positive-low (P-L) pictures in the stress and control conditions. Error bars represent standard error of mean (S.E.M.) (* *p* < 0.001).

**Figure 5 brainsci-10-00995-f005:**
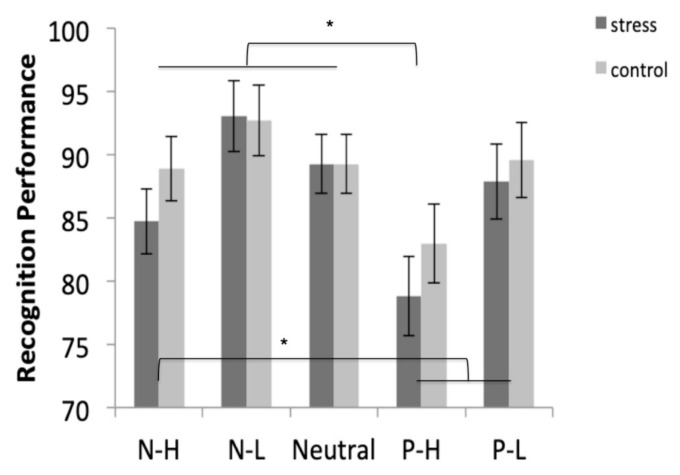
Recognition performance for negative-high (N-H), negative-low (N-L), neutral, positive-high (P-H), and positive-low (P-L) pictures in stress and control conditions. Error bars represent standard error of mean (S.E.M.) (* *p* < 0.05).

**Figure 6 brainsci-10-00995-f006:**
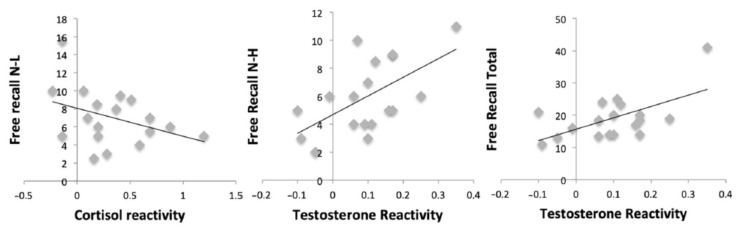
Scatterplots between hormonal reactivity to the Maastricht Acute Stress Test (MAST) and free recall performance.

**Table 1 brainsci-10-00995-t001:** Summary of mixed ANOVAs.

**Response**	**Condition**	**Time**	**Condition × Time**
**Anxiety Self-Reported**	F _(1, 33)_ = 22.23, *p* < 0.001, η^2^_p_ = 0.402	F _(1, 33)_ = 25.88, *p* < 0.001, η^2^_p_ = 0.440	F _(1, 33)_ = 31.67, *p* < 0.001, η^2^_p_ = 0.490
**Cortisol**			
Acquisition Session	F _(1, 35)_ = 19.944, *p* < 0.001, η^2^_p_ = 0.363	F _(1.745, 61.084)_ = 6.474, *p* = 0.004, η^2^_p_ = 0.156	F _(1.745, 61.084)_ = 15.970, *p* < 0.001, η^2^_p_ = 0.313
Memory Testing Session	F _(1, 35)_ = 2.163, *p* = 0.150, η^2^_p_ = 0.058	F _(1, 35)_ = 14.170, *p* = 0.001, η^2^_p_ = 0.288	F _(1, 35)_ = 0.001, *p* = 0.975, η^2^_p_ = 0.001
**Testosterone**			
Acquisition Session	F _(1, 35)_ = 0.275, *p* = 0.604, η^2^_p_ = 0.008	F _(2.375, 83.131)_ = 3.747, *p* = 0.021, η^2^_p_ = 0.097	F _(2.375, 83.131)_ = 2.101, *p* = 0.120, η^2^_p_ = 0.057
Memory Testing Session	F _(1, 35)_ = 0.135, *p* = 0.715, η^2^_p_ = 0.004	F _(1, 35)_ = 0.038, *p* = 0.847, η^2^_p_ = 0.001	F _(1, 35)_ = 0.214, *p* = 0.646, η^2^_p_ = 0.006
**Memory**	**Condition**	**Category**	**Condition × Category**
**Ratings Pictures**			
Valence	F _(1, 35)_ = 2.366, *p* = 0.133, η^2^_p_ = 0.063	F _(2.067, 72.353)_ = 206.301, *p* < 0.001, η^2^_p_ = 0.855	F _(2.067, 72.353)_ = 0.969, *p* = 0.387, η^2^_p_ = 0.027
Arousal	F _(1, 35)_ = 0.331, *p* = 0.560, η^2^_p_ = 0.009	F _(2.787, 97.532)_ = 61.150, *p* < 0.001, η^2^_p_ = 0.636	F _(2.787, 97.532)_ = 0.413, *p* = 0.729, η^2^_p_ = 0.012
**Memory Performance**			
Free Recall	F _(1, 35)_ = 0.000, *p* = 0.992, η^2^_p_ = 0.000	F _(4, 140)_ = 23.400, *p* < 0.001, η^2^_p_ = 0.401	F _(4, 140)_ = 1.042, *p* = 0.388, η^2^_p_ = 0.029
Recognition	F _(1, 34)_ = 0.485, *p* = 0.491, η^2^_p_ = 0.014	F _(2.758, 83.764)_ = 8.045, *p* < 0.001, η^2^_p_ = 0.191	F _(2.758, 83.764)_ = 0.493, *p* = 0.672, η^2^_p_ = 0.014

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
