# Peer review of "Post-Encoding Stress Does Not Enhance Memory Consolidation: The Role of Cortisol and Testosterone Reactivity"

_brainsci, 2020, doi:10.3390/brainsci10120995_

Round 1

Reviewer 1 Report

In this manuscript, Hidalgo and colleagues describe the results of a study aimed at examining the effects of the MAST on post-encoding stress, as well as examining associations between memory and both cortisol and testosterone. Although the study methods are generally good (though the choice to include only women taking hormonal contraceptives is a bit odd given the meta-analysis finding that the authors cite) and the manuscript is extremely well written, the study is critically limited by its very small sample size and corresponding lack of statistical power. Because women on contraceptives show no beneficial effect of post-encoding stress on memory, the effective sample size for this study is 8 in the stress group and 13 in the control group. This very small sample size precludes definitive inferences about any of the authors’ primary analyses, which substantially tempers my enthusiasm for this paper. I describe my concerns in detail below.

Major concerns:

  1. The major concern that I have with this study is that it is simply too underpowered to detect any effects. A recent meta-analysis found that women taking hormonal contraceptives showed no effect of post-encoding stress on memory (Shields et al., 2017), and the authors only had 31% power to detect in men what that meta-analysis states is the expected effect (g = .69) of post-encoding stress on memory under the best experimental conditions. Because of that, it is not surprising to me that the authors did not find an effect: There is good reason to expect that the effect does not exist in their female participants, and the authors were twice as likely to fail to find a significant effect as they were to find a significant effect—given their sample size and expected effect size—in their male participants. Because of that, I do not believe any definitive inferences about the effects of stress on memory can be drawn from the results of this study. The authors should, at the very least, replace the note about including women on contraceptives not explaining the null effect on men in their study with a note about how they may have lacked the statistical power to observe an effect of stress on memory in men. Moreover, they should remove statements about the MAST not enhancing memory consolidation (e.g., in their conclusion); it is very possible that it does enhance consolidation and the authors simply lacked the statistical power to detect that effect.
  2. Related to the above, the small sample size can also explain the failure to find testosterone changes in response to stress. The effects of stress on sex hormones are much smaller than the effects of stress on cortisol, and there certainly seems to be an increase from -5 to +10min in men (and maybe women) in their sample (though I do note that men in the control condition also numerically increased over the same window), but they lack the statistical power to detect that. I would caution the authors against arguing that stress does not increase testosterone in this paradigm and suggest that they instead interpret their results in light of the low statistical power.
  3. Related to the small sample size issue, a small sample sizes increases the likelihood of obtaining spurious associations. The authors tested 24 different associations between hormones (two) and memory (six measures of recall [i.e., total, and each of the emotion categories], and six of recognition), and three of those associations were significant. Three out of 24 is 12.5%, which is not much different from the 5% that would be expected by chance alone. I am especially suspicious of the testosterone reactivity associations considering that testosterone did not change overall. Why would testosterone reactivity predict memory if testosterone did not change? What happens when post-manipulation testosterone is used as the predictor rather than testosterone reactivity? And, do any of these associations survive FDR correction?

Minor comments:

  1. The stress and control conditions are highly unbalanced with respect to sex; there are two more women than men in the stress condition, but seven less women than men in the control condition. Do the authors think this may have influenced their results? Can the authors report the means/SDs for memory for each sex in a supplemental file?
  2. Did participants in this study change contexts in any way between learning and stress (e.g., rooms, experimenters, computer monitors, etc.)? The discussion states that it was the same context, but the MAST requires a number of things that are often easier to set up in another room (e.g., a video camera, a cold pressor machine). Was all of the equipment for the MAST already in the room when the participants sat down, or was any of it brought in? Or, did participants change rooms? If the context was exactly the same, this should be stated explicitly within the method section in the procedure.
  3. It would be informative to include scatterplots of the hormone reactivity and memory associations, considering that these associations were a notable result.
  4. Shields et al. (2019, Brain, Behavior, and Immunity) also found beneficial effects of stress on recognition, but only in recollection (similar to the cited Sazma study). Together with the Sazma study, these studies suggest that post-encoding stress may only benefit recollection in recognition memory. Consider mentioning that as a possibility when explaining the nonsignificant effects of stress on recognition in this study, given that recollection was not quantified here.

Reviewer 2 Report

The article: Post-Encoding Stress does not Enhance Memory Consolidation: The Role of Cortisol and Testosterone Reactivity is interesting, but some improvement are necessary.

First of all I suggest to reduce the chapter Introduction and Discussion focusing on the basic concept to make the article easier to the readers.

Also methodology could be restricted.

A table summarizing the date of the subjects it would be useful. Some factors as BMI, nutrition, physical activity, etc. can influence testosterone and cortisol production.

Some concept are expressed without references:

Line 39,41,42, 63.

Line 63: the role of testosterone should be better explained.

Line374-376 is a repetition (see introduction).

Line 378 need a reference. Mechanisms that increases testosterone level should be better explained.

Line 395 Correct the sentence.

Reviewer 3 Report

The study revolves around post-encoding stress and whether it enhances memory consolidation and studies the role of cortisol and testosterone. Authors designed a between-subject study with two sessions (acquisition and memory testing). The acquisition session included either MAST stress task or control (i.e. no stress), while the memory testing consisted of free recall and recognition task. Authors conclude that MAST stressor is able to evoke a psychological and physiological response, however, it does not affect testosterone levels and memory consolidation.

General comments:

  • can authors comment on why did they choose a between-subject design and not within-subject design with more sessions?
  • almost the whole results section is just enumerating results of ANOVAs and posthoc tests and this makes this section very hard to read. I would suggest put results in tables and comment only on significant results in the main text
  • all figures should include stars when the comparison is significant - many people just skim through the figures and now they cannot see what is happening
  • is the number of participants enough for this study? I cannot see any study power calculation
  • English changes are necessary

Specific comments:

  • lines 69-73: First sentence states that stressors like TSST elicit testosterone increase, but the next sentence states that no significant testosterone changes have been reported
  • line 118: is that a raw Chi-square test? Or did you use Yates correction? My quick calculation yields different chi2 and p- values.
  • line 144: STAI abbreviation is not defined here, but later in the text
  • lines 178-179: interrater agreement: what exactly means that agreement was 92.7%? What about e.g. Cohen's kappa?
  • lines 183 - 185: Why was this measure for recognition outcomes chosen? Why not some more classical classification measures as recall or F1 score?
  • lines 207-209: why are some data missing? only hormone concentrations were not normal? Or some other variables? Which variables have been logarithmically transformed? Only the imputed ones? All hormone variables? All study variables? How was the non-normality estimated? Using a Shapiro-Wilk test?
  • lines 212-217: ANOVA with both within- and between-subject variables are usually termed as mixed or split-plot and not repeated measures. Also, this whole sentence is almost unreadable. I would suggest rewriting.
  • line 225: which posthoc test was used?
  • lines 282 and 284: Based on your definition of significance (p < 0.05) all of these were not significant. Not marginally significant, just not significant.
  • line 285: Since the condition factor was not significant, this sentence is not true.
  • lines 306 and 307: twice positive-hight
  • lines 309-310: again, not significant, please do not use phrases like "significant as a trend", it's simply not significant
  • lines 409-410: this is interesting, do you think it is conceivable to design a study with multiple expected cortisol levels, such that this curvilinear relationship can be revealed? 

In general, my main problem is with the presentation of the results and some methods used - I suggest the authors add tables for ANOVA and posthoc results as those are much more readable than dense text. Also, the significance should be pinpointed in the figures. Also, I want to say I find the introduction nicely written, stating the current knowledge in the field and clearly presented hypotheses for this study.

All in all, I suggested reworking the results section and make some comments on the methodology such that it is crystal clear and submit after revision.

Reviewer 4 Report

The authors tested salivary cortisol and testosterone reactivity in randomly assigned case-control healthy adults in response to the Maastricht Acute Stress Test (MAST), as well as the association between acute stress and the recall and recognition of images. While the authors found that the MAST elicited a cortisol response and self-reported anxiety, neither testosterone levels nor memory consolidation were influenced by the stress test.

Overall the writing is clear and the topic is of interest to readers. The use of a control group improves interpretation of the findings. Comments and questions for the authors are below:

  1. Can the authors confirm there were no sex differences in group assignments, stress vs. control? How were the participants randomly assigned to condition, and was an attempt made to balance the sexes?
  2. The authors state that all women were using hormonal contraceptives - was there any assessment of which phase of the menstrual cycle women were in during the test?
  3. The sample sizes per group are relatively small, could the authors speak to design of the study and power considerations?
  4. On page 4, line 173 - what is meant by "recommendations?" Do the authors mean the experimenter checked that participants did not eat, drink coffee etc. before the study visit?
  5. On page 5 of the methods, it seems the saliva collection surrounding the MAST took place at -80 minutes, -10 minutes, +10 minutes, and +25 minutes. Elsewhere in the methods (page 6) and results, it seems this was actually "baseline"/"BL" (when?), -5, +10, +25 minutes. Could the authors please clarify?
  6. Could the authors report why there were missing cortisol and testosterone data for the participants?
  7. Page 8 line 306 and 307: positive-high results are reported twice, with different values. One of these must be positive-low.
  8. Could the authors speak more about the expected scores for the photo recall and recognition tasks? How well did participants perform? The discussion touches on the possibility of a ceiling effect.

Minor comments:

  1. Should reference 8 be "Preuss" and Wolf in the text and references?
  2. I would suggest changing the word "response" on page 5, lines 231 and 238, to indicate that this is self-reported.
  3. Page 8 line 305 and 318, and page 9 line 329 - I believe the first word/phrase is meant to be a further sub-heading.

Round 2

Reviewer 3 Report

First of all, I would like to thanks authors for taking the time and providing us with a significantly improved version of the manuscript. Also, I want to point out that the rebuttal letter is nicely written and answers to all reviewers' comments.

Major comment: Again on the study power. On my request on computing the study power, the authors added a paragraph reporting Cohen's d (lines 226-226). This is, however, only the estimate of the effect size. For power analysis, I meant reporting the power in the sense of 1 - β with β being the type II error probability. This can be also done in G*Power. The posthoc achieved power can be computed as follows in G*Power:

  • test family: F-tests
  • statistical test: ANOVA: repeated measures, within-between interaction
  • type of analysis: posthoc
  • effect size f: can be determined in the G*Power from partial η2, which you are reporting anyway
  • α, total sample size, number of groups and others are clear from the study design
  • then the power calculation appears in the lower left, (e.g. when I insert some of your values for testosterone as the response variable for a time as a factor, assuming within measures correlation of 0.4 and sphericity correction of ε=0.8 I am getting a power of 98.5%)
  • I would suggest to compute statistical powers for their ANOVAs and then report a minimum value in significant cases  

Minor comments: 

  • authors mention Bonferroni correction (in the response to reviewer 1). If I am not mistaken I do believe that Bonferroni procedure controls FWER and might be too conservative/stringent for this case. I would rather correct for multiple comparisons controlling the FDR, e.g. using Benjamini–Yekutieli procedure. This is a mere side note since I am not sure if or how this would change the survival rate of reported correlations.
  • after power calculations, as I suggest, it would be easy for authors to compute the required sample size a priori given the expected effect size and report the number of participants to achieve greater power in the conclusions (line 438)

All in all, I would really like to see a proper power calculation, but otherwise, I like where this is going.
